

# Identification and validation of a three-gene signature as a candidate prognostic biomarker for lower grade glioma

Kai Xiao, Qing Liu, Gang Peng, Jun Su, Chao-Ying Qin and Xiang-Yu Wang

Department of Neurosurgery, Xiangya Hospital of Central South University, Changsha, Hunan, China

## ABSTRACT

**Background:** Lower grade glioma (LGG) are a heterogeneous tumor that may develop into high-grade malignant glioma seriously shortens patient survival time. The clinical prognostic biomarker of lower-grade glioma is still lacking. The aim of our study is to explore novel biomarkers for LGG that contribute to distinguish potential malignancy in low-grade glioma, to guide clinical adoption of more rational and effective treatments.

**Methods:** The RNA-seq data for LGG was downloaded from UCSC Xena and the Chinese Glioma Genome Atlas (CGGA). By a robust likelihood-based survival model, least absolute shrinkage and selection operator regression and multivariate Cox regression analysis, we developed a three-gene signature and established a risk score to predict the prognosis of patient with LGG. The three-gene signature was an independent survival predictor compared to other clinical parameters. Based on the signature related risk score system, stratified survival analysis was performed in patients with different age group, gender and pathologic grade. The prognostic signature was validated in the CGGA dataset. Finally, weighted gene co-expression network analysis (WGCNA) was carried out to find the co-expression genes related to the member of the signature and enrichment analysis of the Gene Ontology (GO) and the Kyoto Encyclopedia of Genes and Genomes (KEGG) pathway were conducted for those co-expression network. To prove the efficiency of the model, time-dependent receiver operating characteristic curves of our model and other models are constructed.

**Results:** In this study, a three-gene signature (WEE1, CRTAC1, SEMA4G) was constructed. Based on the model, the risk score of each patient was calculated with LGG (low-risk vs. high-risk, hazard ratio (HR) = 0.198 (95% CI [0.120–0.325])) and patients in the high-risk group had significantly poorer survival results than those in the low-risk group. Furthermore, the model was validated in the CGGA dataset. Lastly, by WGCNA, we constructed the co-expression network of the three genes and conducted the enrichment of GO and KEGG. Our study identified a three-gene model that showed satisfactory performance in predicting the 1-, 3- and 5-year survival of LGG patients compared to other models and may be a promising independent biomarker of LGG.

Corresponding author
Xiang-Yu Wang,
wxyMGH@gmail.com

## INTRODUCTION

With the development of sequencing and bioinformatics technologies, accumulating studies have revealed that different patients may be similar in glioma grade but differ greatly in molecular characteristics, clinical prognosis and treatment response. So many central nervous system tumors were named according to molecular parameters and histopathologic diagnosis, especially gliomas, ependymomas and medulloblastomas in the 2016 revision of the WHO classification (*Zhang et al., 2019b*). As we know, some molecular markers, such as MGMT (O6-methylguanine DNA methyltransferase) (*Binabaj et al., 2018*), isocitrate dehydrogenase (IDH) (*Kwon et al., 2019*), epidermal growth factor receptor (EGFR) (*Chistiakov, Chekhonin & Chekhonin, 2017*) and phosphatase and tensin homolog (PTEN) (*Koshiyama et al., 2017*) that have contributed to personalized therapeutic approaches and targeted anti-glioblastoma therapies have been routinely tested in glioblastoma patients clinically (*Yin et al., 2019*). However, there are few specific clinical indicators and therapeutic targets for LGGs compared to glioblastoma, so there is an urgent need to elucidate the mechanism of glioma development and progression, which can provide potential treatment targets for LGGs.

In this study, gene expression RNAseq data and corresponding clinical information of LGG patients were downloaded from UCSC Xena (https://xenabrowser.net/hub/) and Chinese Glioma Genome Atlas (CGGA, http://www.cgga.org.cn/). By analyzing data from UCSC Xena using a robust likelihood-based survival model and Cox regression, we developed a three-gene signature that provides effective survival risk stratification of patients with LGG and validated the signature in the CGGA dataset. These results demonstrate the potential of the three-gene signature for survival prediction of patients with LGG and provide new potential molecular treatment targets for LGGs.

## MATERIALS AND METHODS

### Dataset of patients with LGG

The LGGs RNA sequencing (RNAseq) data and corresponding clinical information were downloaded from The Cancer Genome Atlas (TCGA) hub by the University of California, Santa Cruz, Xena browser (https://xenabrowser.net/hub/) and CGGA data portal (http://www.cgga.org.cn/) respectively. The TCGA RNAseq data (level 3) shows the gene-level transcription estimates, as in $\log2(x + 1)$ transformed RSEM normalized count. The CGGA data displays the gene expression level as fragments per kilobase transcriptome per million fragments (FPKM), which has been standardized. Expressed gene defined only if its expressed level is larger than zero at half of samples. Only patients with a clear information of survival and detailed history of radiotherapy and chemotherapy/molecular therapy were included in the study. Finally, 456 cases from TCGA dataset and 159 cases from the CGGA dataset were included in the training set and validation set respectively. Table 1 summarized the clinical characteristics and therapy information of
**Table 1 Clinical parameters of patients in the training set and validation set.**

| Variables | Training set ($n = 456$) | Validation set ($n = 159$) | p-value |
|---|---|---|---|
| **Age group (Median)** | | | 0.5448 |
| Younger | 232 | 86 | |
| Old | 224 | 73 | |
| **Sex** | | | 0.1894 |
| Female | 210 | 63 | |
| Male | 246 | 96 | |
| **Grade** | | | 0.09385 |
| G2 | 221 | 90 | |
| G3 | 235 | 69 | |
| **Molecular therapy** | | | |
| Yes | 263 | / | |
| NO | 193 | / | |
| **Chemoterapy** | | | |
| Yes | / | 80 | |
| NO | / | 79 | |
| **Risk level** | | | 1 |
| High | 228 | 79 | |
| Low | 228 | 80 | |
| **IDH_status** | | | |
| Wildtype | / | 43 | |
| Mutant | / | 116 | |
| **1p19q_status** | | | |
| Non_codel | / | 52 | |
| Codel | / | 107 | |
| **Radiation therapy** | | | 3.52E−10 |
| Yes | 280 | 141 | |
| NO | 176 | 18 | |
| **Age (years)** | | | |
| Mean ± SD | 43.4 ± 13.3 | 40.7 ± 10.9 | 0.989 |
| Median | 41 | 40 | |
| **Vital status** | | | |
| Alive | 341 | 77 | |
| Dead | 115 | 82 | |
| **Survival time (days)** | | | |
| Mean | 998.6 ± 953.8 | 2,024.7 ± 1334.3 | 5.27E−24 |
| Median | 714.5 | 2,340 | |
| **Histologic type** | | | |
| Astrocytoma | 162 | 34 | |
| Oligodendroglioma | 171 | 21 | |
| Oligoastrocytoma | 123 | 35 | |
| Anaplastic astrocytomas | / | 26 | |
| Anaplastic oligoastrocytomas | / | 32 | |
| Anaplastic oligodendrogliomas | / | 11 | |
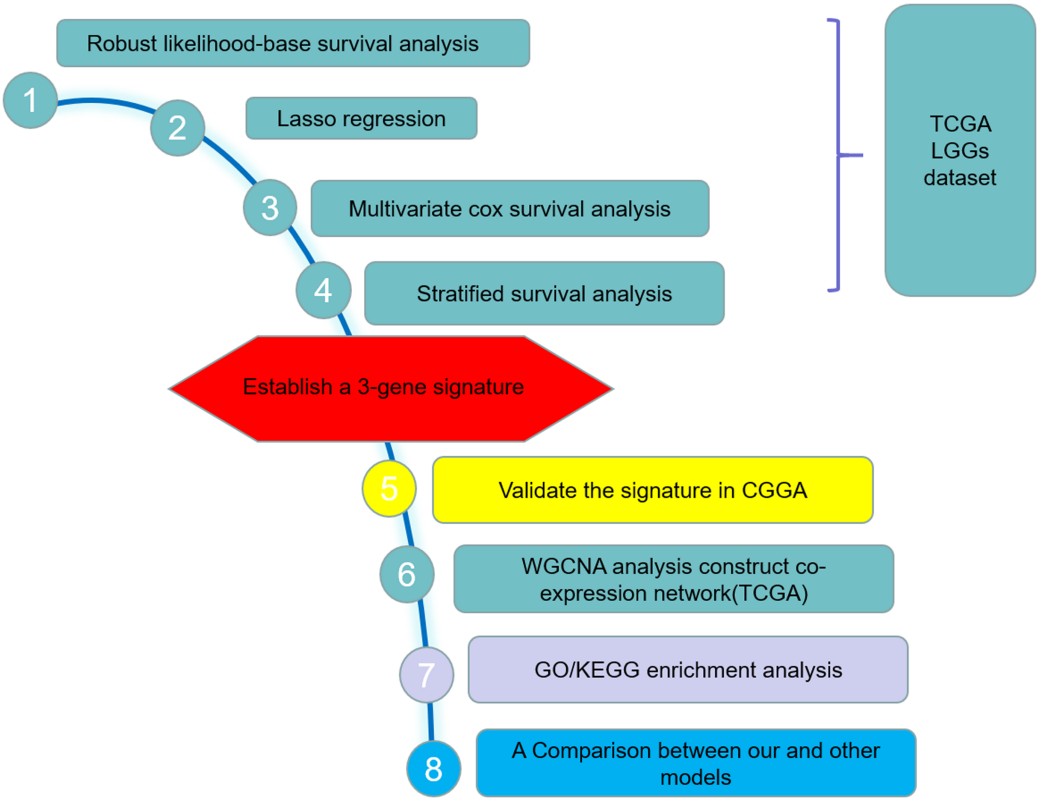

**Figure 1 Study outline.** The outline indicates the exploration process.

the training set and validation set. The workflow presentation of this study is shown in Fig. 1.

## Identification of survival-related genes and construction of the prognostic model

By using the rbsurv package in R, a robust likelihood-based survival model was conducted to identify survival-related genes (*Cho et al., 2009*). The rbsurv package is a software program, which selects survival-associated genes based on the partial likelihood of the Cox model and adopts a cross-validation approach for robustness. According to the description of the rbsurv package, prior gene selection such as univariate survival modeling can be performed if necessary and the univariate survival modeling can be performed in this software program. Compared to the survival modeling without an adjustment of risk factors, the robust likelihood-based survival model can improve the ability to discover truly survival-associated genes by modeling genes after adjusting for certain risk factor. Thus, we directly conduct a robust likelihood-based survival model to screen for the prognostic genes. The robustness test was performed on 20,530 genes and 456 samples. After 10 iterations, 29 prognostic related genes were selected. With the help of glmnet and survival package in R, least absolute shrinkage and selection operator (LASSO) regression and the multivariable Cox proportional hazard regression method were used to further identify

the survival-related prognostic model. The same approach was used to identify gene signatures for endometrial carcinoma (*Ouyang et al., 2019*). At last, three prognostic survival-related genes that were independent survival predictors and their regression coefficients were obtained at a threshold of $p < 0.05$. Based on the median expression value of each survival-related gene, we dichotomized 456 LGGs patients into low and high expression groups and compare the survival rate between the two groups by Kaplan–Meier plots and Log-rank test. According to the estimated regression coefficients, a prognostic risk score for each patients was then calculated (*Wang et al., 2019*). The risk score = (0.4470 × expression level of WEE1) + (−0.1530 × expression level of CRTAC1) + (−0.3723 × expression level of SEMA4G). With the three-gene signature, 456 LGGs patients were divided into high-risk and low-risk groups with the median risk score as the cut-off value. Kaplan–Meier curves were performed to estimate and compare the survival for TCGA LGGs patients with a high score or a low score. The receiver operating characteristic (ROC) curve and area under the curve (AUC) were applied to evaluate the prediction accuracy of the risk score model. Furthermore stratified survival analysis was performed in patients with different age group (younger, old), gender (male, female) and pathologic grade (G2, G3).

Univariate and multivariate Cox hazard regression analysis were conducted for the potential prognostic factors such as age group (younger vs. old), gender (male vs. female), pathologic grade (G2 vs. G3), radiation therapy (Yes vs. No), molecular therapy (Yes vs. No) and risk score (High vs. Low).

## Validation of the prognostic model in the CGGA

The prognostic model was validated in the CGGA mRNAseq_325 cohort. Only patients with a clear information of survival, detailed history of radiotherapy and chemotherapy were included in the study. Finally, 159 cases from the CGGA mRNAseq_325 cohort were included in the validation set.

## Exploring co-expression genes by WGCNA

To explore the regulatory network of the three genes, Gene Co-expression Network Analysis (WGCNA) was performed in training set by the R package WGCNA (*Langfelder & Horvath, 2008*). The top 50% variance of genes were selected for WGCNA. In other words, WGCNA based on 456 samples and 10,256 genes. First, RNAseq data were filtered to reduce outliers. Using the absolute value of the correlation between the expression levels of transcripts, a co-expression similarity matrix was constructed. Then, the co-expression similarity matrix was transformed to the adjacency matrix by choosing nine as a soft threshold. Co-expression gene module was established by the topological overlap measure. In order to identify the significance of each module, gene significance (GS) was calculated to estimate the correlation between genes and sample traits. Module significance (MS) was defined as the average GS within modules and was calculated to measure the correlation between modules and sample traits (vital status). Finally, the "vital status" related modules that contain the three genes as members and genes belong to such modules were identified. Genes interacted with those three genes
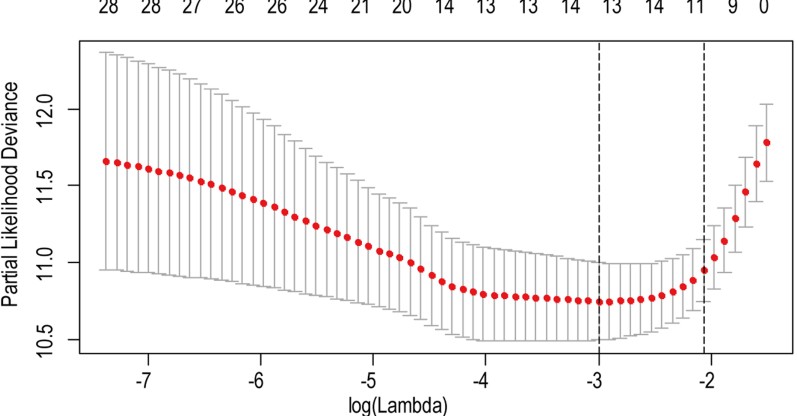

**Figure 2 The LASSO regression used to reduce the dimensionality of survival related genes.**

were screened and the co-expression network was constructed by Cytoscape software (*Shannon et al., 2003*).

## Functional enrichment analysis

Using Enrichment analysis of Gene Ontology (GO) and Kyoto Encyclopedia of Genes and Genomes (KEGG) pathway were conducted via the clusterProfiler package in R language (*Yu et al., 2012*) for those genes that belong to the "vital status" related modules associated with the three genes. Benjamini–Hochberg (BH)-adjusted *p*-value < 0.05 were considered significant.

## RESULTS

### Three prognostic genes were identified in TCGA dataset and validated in CGGA dataset

A total of 456 patients and 20,530 genes were included in the TCGA-LGG to train the prognostic model. The robust likelihood-based survival model found 29 survival-related genes, 13 genes were obtained through LASSO Cox method (Fig. 2). We further reduced the dimensionality of these high-dimensional data by multivariate Cox proportional hazard regression model. Finally, three genes that were independent survival predictors were identified as survival prediction signature. Those three genes included in the model were WEE1, SEMA4G, CRTAC1. It has been reported that WEE1 is closely related to the growth, invasion and migration of glioma (*Wu et al., 2019*). Currently, there is no study revealing the role of SEMA4G and CRTAC1 in gliomas. After calculating the risk score, patients were divided into a high- and low-risk group based on the median cut-off point of the risk score. The three-gene signature risk score distribution is shown in Fig. 3A. Besides, the relationship between risk score and the status of the LGGs was calculated (Fig. 3B). As shown in the heat map of the Fig. 3C, a remarkable high expression was noted for WEE1 in the high-risk group, while a lower expression was observed for the other genes in the high-risk group (Fig. 3C). Patients in the high-risk group were significantly worse off the overall survival time compared to the low-risk group

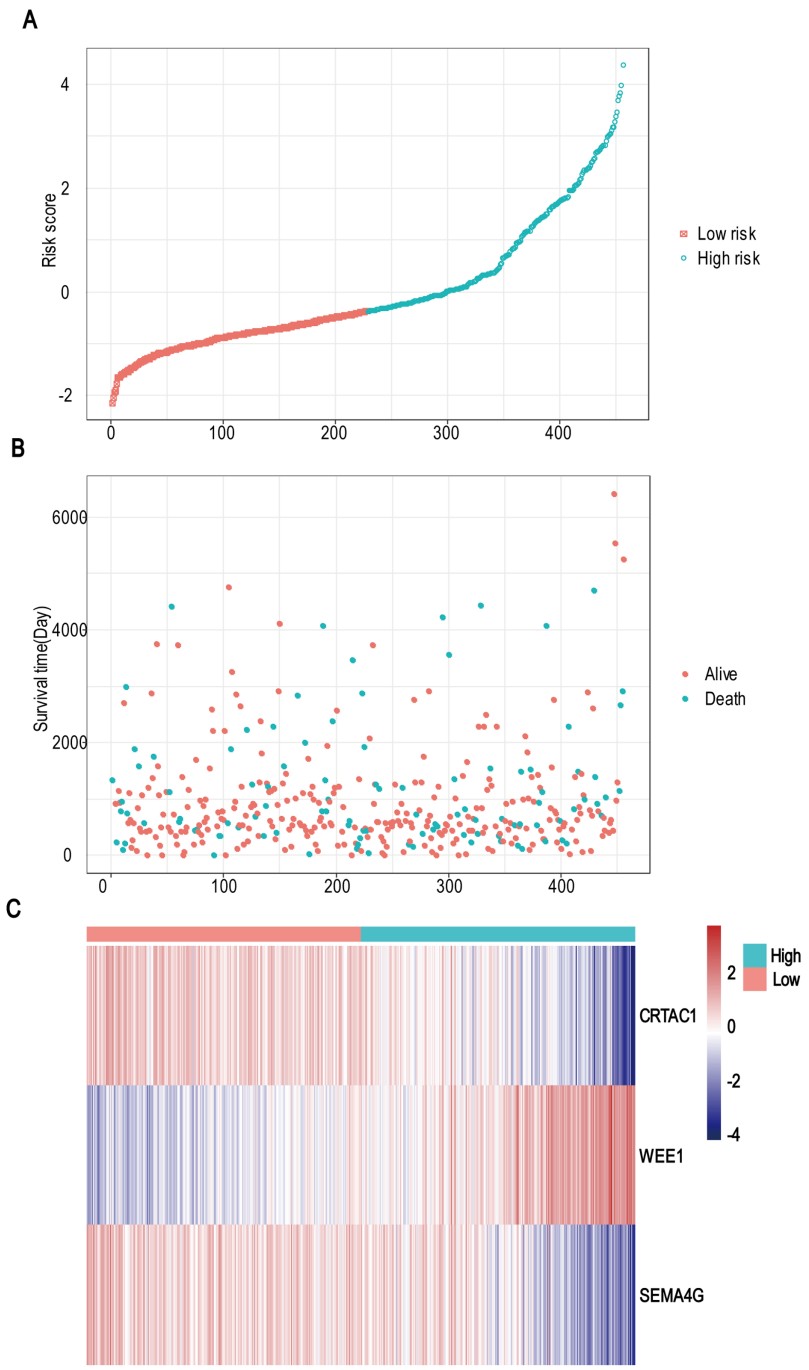

**Figure 3 Risk score analysis, survival status and survival time between two risk group and expression distribution of the three-gene signature in TCGA dataset.** (A) The three-gene signature risk score distribution. (B) Scatterplot of patient survival status ordered by risk score. (C) The heat-map of the three-gene expression profiles after standardized and centralized.

($p < 0.0001$) (Fig. 4A). The area under ROC curve of the signature for 1-, 3- and 5-year overall survival was 0.908, 0.878 and 0.827, respectively, in training set. (Fig. 4B). A similar result can be noted in the validation dataset (Fig. 4C). The area under ROC curve of the

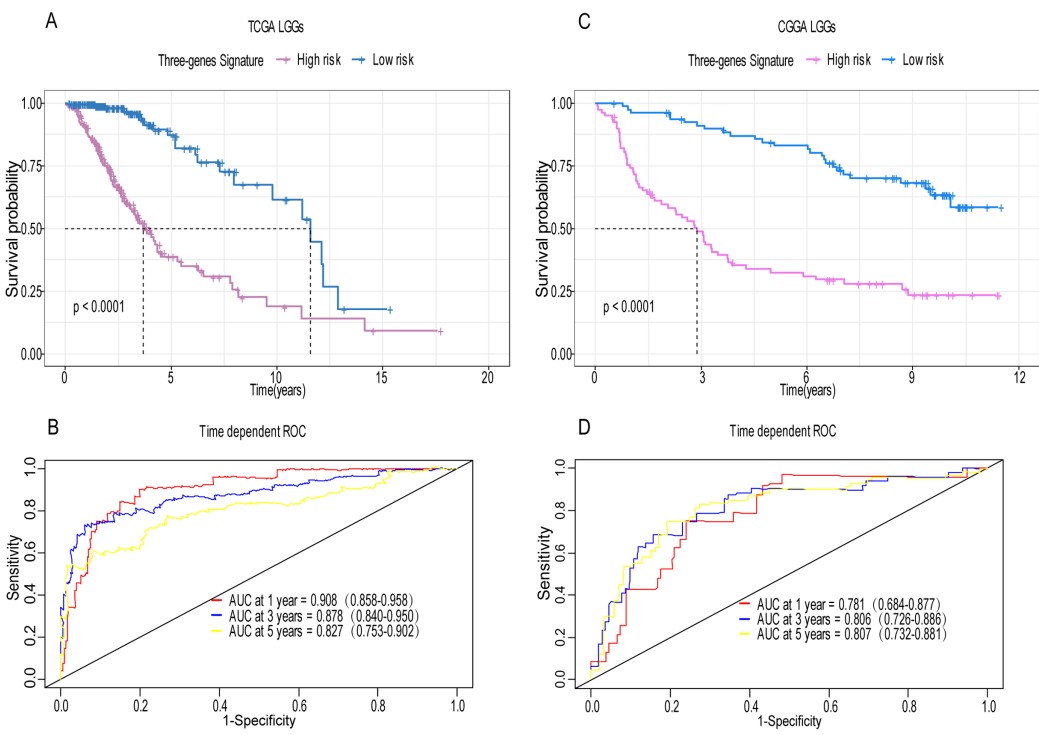

**Figure 4 Establishment and verification of the risk model in the training set and validation set.**
(A) Patient in high-risk group displayed significantly shorter survival time compared to those in low-risk group in training set ($p < 0.0001$). (B) The ROC for predicting the 1-, 3- and 5-year survival and AUC for the risk score model showed good accuracy in training set, the area under the ROC curve were 0.908 (95% confidence interval (CI) [0.858–0.958]), 0.878 (95%CI [0.840–0.950]) and 0.827 (95%CI [0.753–0.902]). (C, D) The same result can be observed in the validation set, and the area under the ROC curve were 0.781 (95%CI [0.684–0.877]), 0.806 (95%CI [0.726–0.886]) and 0.807 (95%CI [0.732–0.881]) for 1-, 3- and 5-year survival.                

signature for 1-, 3- and 5-year overall survival was 0.781, 0.806 and 0.807, respectively, in validation set. (Fig. 4D). Moreover, the predicting power of the risk score model was not decreased in subgroup analysis for age group (younger, $p = 0.00012$; old, $p < 0.0001$), gender (male, $p < 0.0001$; female, $p < 0.0001$), and pathologic grade (G2, $p = 0.00013$; G3, $p < 0.0001$) in the training set (Figs. 5A–5F). The same trend can be observed in the validation dataset (Figs. 6A–6F). For the WEE1, the member of high expression group had significantly shorter survival than those in low expression group ($p < 0.0001$) (Fig. 7A). For the SEMA4G and CRTAC1, the member of high expression group had significantly longer survival than those in low expression group ($p < 0.0001$) (Figs. 7B and 7C). The expression level of WEE1 was significantly higher in grade III compared to grade II ($p < 0.0001$), while the other are opposite (Fig. 8A). These results can also be verified in the validation dataset (Figs. 7D–7F and 8B).

Multivariate Cox proportional hazard regression demonstrated that age group (HR = 0.274, $p = 2.21E−09$), pathologic grade (HR = 2.49, $p = 0.00011$) and risk score (HR = 0.198, $p < 0.000000000168$) were independent prognostic factors in the training set, while pathologic grade (HR = 3.799, $p = 0.00000151$), 1p19q status (HR = 4.566,

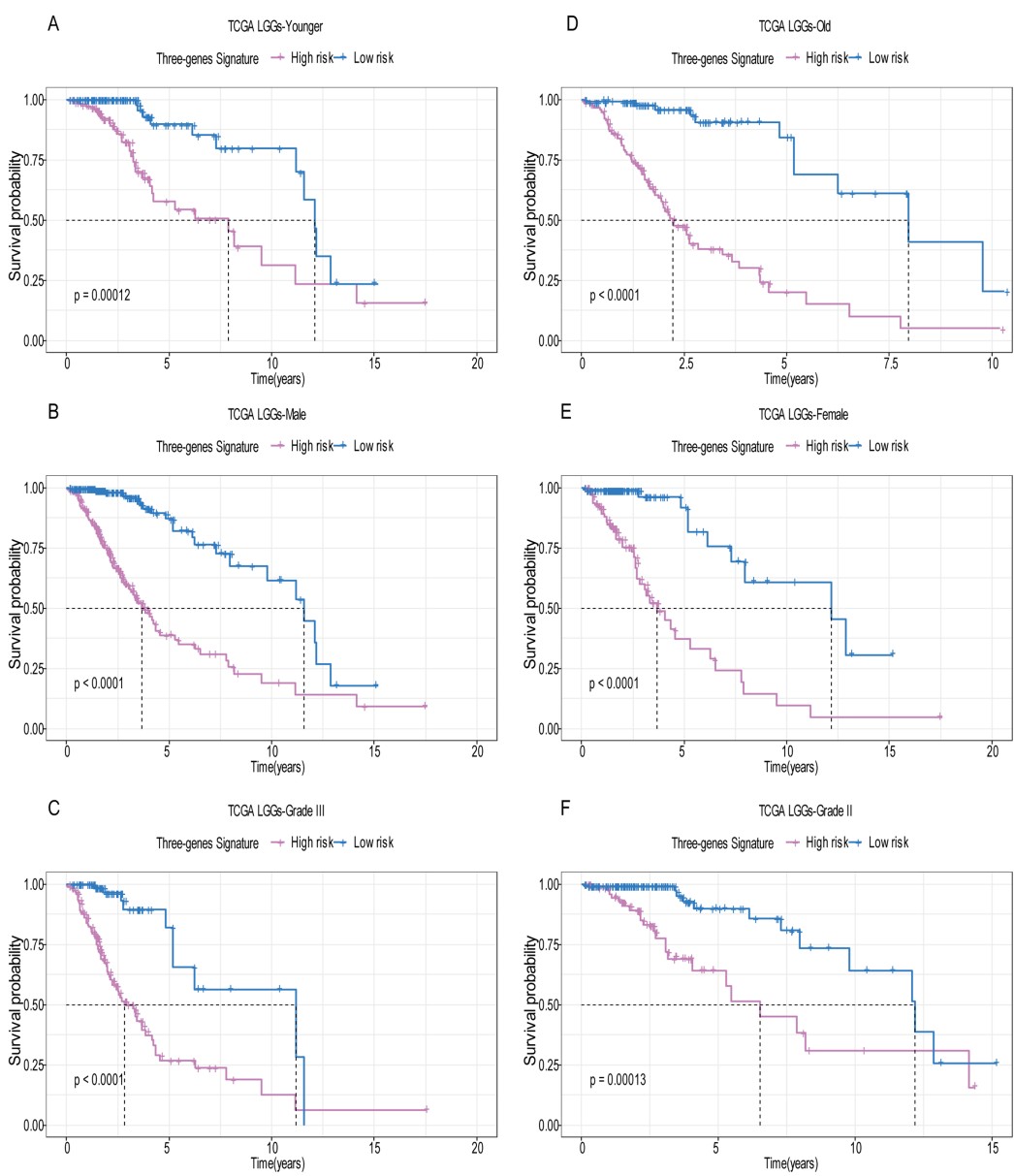

**Figure 5 Stratified survival analysis based on the risk model in the training set.** Based on the risk score model, stratified survival analysis performed in patients with different age group (A and D), gender (B and E), and pathologic grade (C and F) in the training set.

$p = 0.0000388$), radiation therapy (HR = 0.524, $p = 0.046$), and risk score (HR = 0.415, $p = 0.000653$) were independent prognostic factors in validation dataset (Table 2).

## Calculation of module-trait correlation in LGGs and module visualization of the network connections

Using the R package WGCNA, gene modules were identified based on the top 50% variance of genes. To analyze the relationship between gene modules and sample clinical information, we used the module eigengene (ME) as the overall gene expression level of

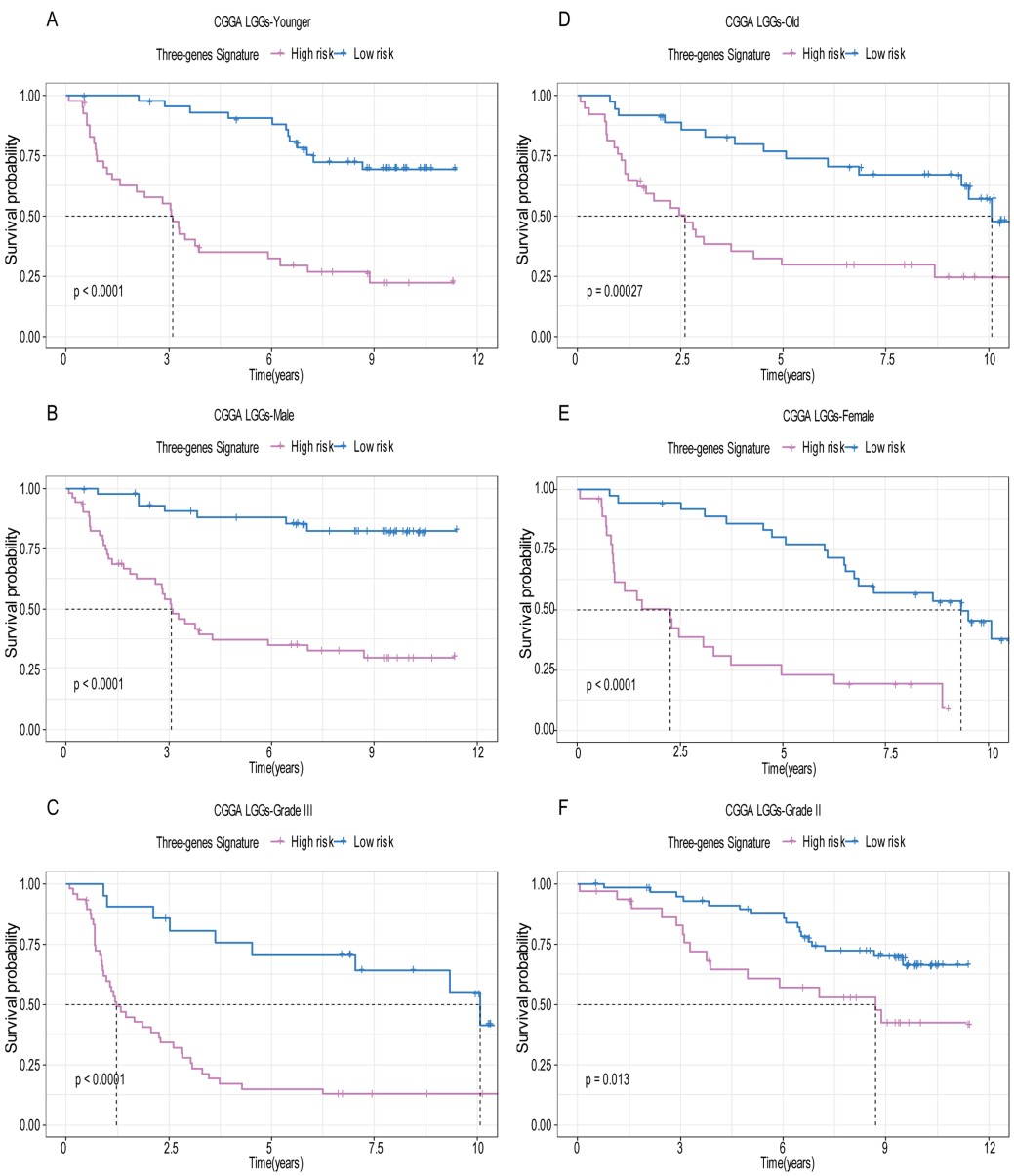

**Figure 6 Stratified survival analysis based on the risk model in the validation set.** Based on the risk score model, stratified survival analysis performed in patients with different age group (A and D), gender (B and E), and pathologic grade (C and F) in the validation set.

the corresponding modules and calculated correlations with clinical phenotypes, for example, vital status. We obtained 16 gene modules (Figs. S1A–S1D) with size ranging from 31 to 1,501 genes. We assigned each co-expression module an arbitrary color for reference: black, blue, brown, cyan, green, greenyellow, lightcyan, magenta, midnightblue, pink, purple, red, salmon, tan, turquoise and yellow. These modules contained 449, 1,352, 850, 46, 519, 91, 31, 201, 43, 336, 135, 462, 51, 90, 1,501 and 845 genes, respectively. As a single group, the non-co-expressed group designated as "grey" based on the WGCNA

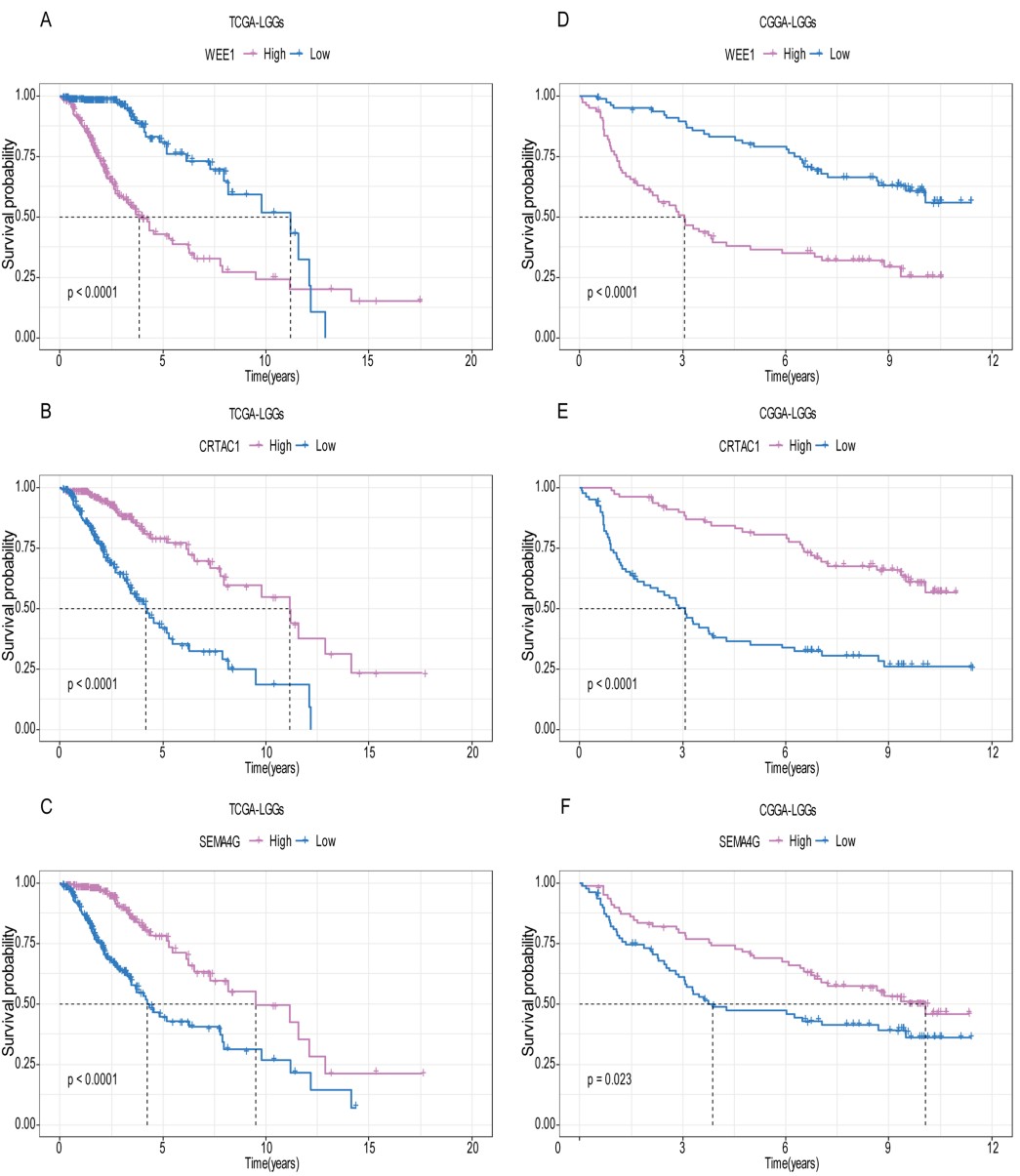

**Figure 7 The expression level of the three genes can divided the patients into different prognostic group in both training set and validation set.** For the WEE1, the member of high expression group had significantly shorter survival than those in low expression group ($p < 0.0001$) (A and D). For the CRTAC1, the member of high expression group had significantly longer survival than those in low expression group ($p < 0.0001$) (B and E). For the SEMA4G, the member of high expression group had significantly longer survival than those in low expression group ($p < 0.0001$) (C and F).

developer's rationale. Vital status related modules, such as yellow, green, black modules that contain the three genes as members and genes belong to such modules were screened (Fig. S1D). Finally, 32 genes were discovered to be co-expressed with CRTAC1, 181 genes were co-expressed with WEE1, six genes with SEMA4G. We exported the screened genes and three prognostic survival-related genes into Cytoscape and constructed the co-expression network (Fig. 9).

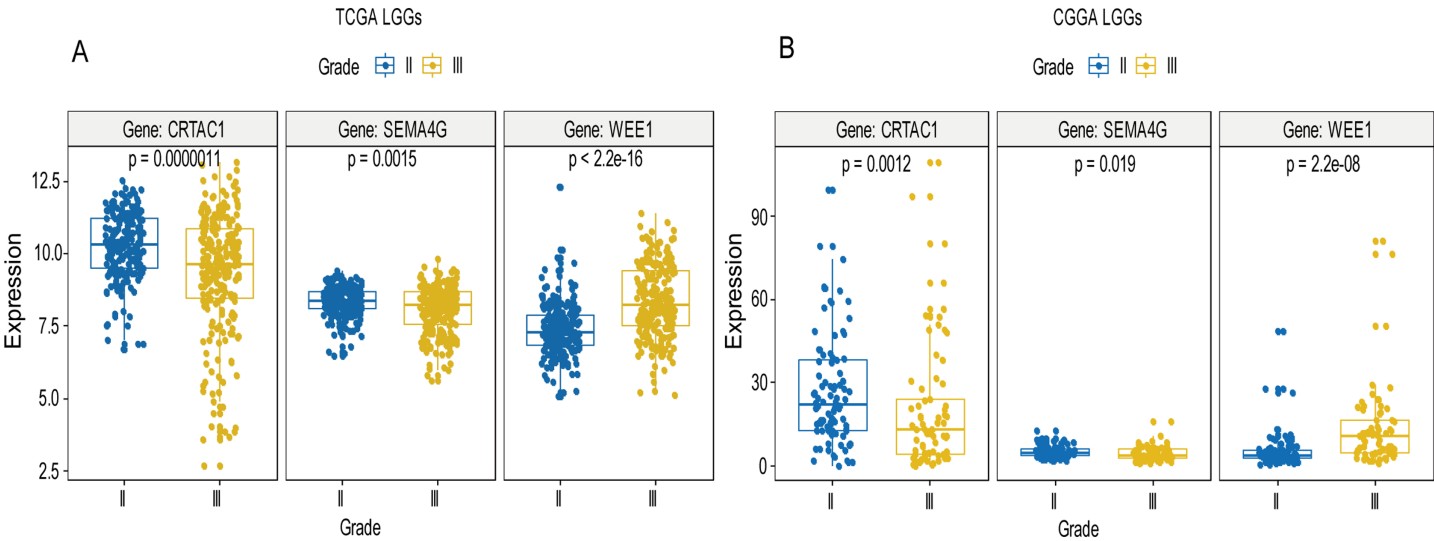

**Figure 8 Expression of the three genes between grade II tumor and grade III tumor in training set and validation set.** In the training set, the expression level of WEE1 was significantly higher in grade III compared to grade II ($p < 0.0001$), while the other are opposite (A). These results can also be verified in the validation dataset (B).

**Table 2 Univariate and multivariate Cox regression analyses of the risk score and other clinicopathological factors in training set and validation set.** HR, Hazard ratio; 95%CI, 95% confidence interval.

| Variables | | Training set ($n$ = 456) | | | | | | Validation set ($n$ = 159) | | | | | |
|---|---|---|---|---|---|---|---|---|---|---|---|---|---|
| | | Univariate | | | Multivariate | | | Univariate | | | Multivariate | | |
| | | HR | 95%CI | $p$-value | HR | 95%CI | $p$-value | HR | 95%CI | $p$-value | HR | 95%CI | $p$-value |
| Age group (Median) | Younger vs. old | 0.278 | 0.184–0.420 | 1.3E−09 | 0.274 | 0.179–0.418 | 2.21E−09 | 0.817 | 0.530–1.261 | 0.363 | 1.051 | 0.654–1.691 | 0.837 |
| Sex | Male vs. Female | 1.043 | 0.721–1.509 | 0.823 | 1.081 | 0.743–1.572 | 0.686 | 0.641 | 0.416–0.989 | 0.044 | 0.651 | 0.412–1.028 | 0.066 |
| Grade | G3 vs. G2 | 3.301 | 2.196–4.963 | 9.3E−09 | 2.49 | 1.568–3.953 | 0.00011 | 3.590 | 2.292–5.625 | 2.4E−08 | 3.799 | 2.205–6.545 | 0.00000151 |
| Molecular therapy | Yes vs. No | 1.366 | 0.924–2.018 | 0.117 | 0.893 | 0.578–1.379 | 0.608 | / | / | / | / | / | / |
| Chemoterapy | Yes vs. No | / | / | / | / | / | / | 2.216 | 1.409–3.485 | 0.00057 | 1.041 | 0.614–1.765 | 0.881 |
| Risk level | Low vs. High | 0.188 | 0.118–0.299 | 2.1E−12 | 0.198 | 0.120–0.325 | 1.68E−10 | 0.246 | 0.153–0.394 | 5.7E−09 | 0.415 | 0.251–0.688 | 0.000653 |
| IDH_status | Wildtype vs. Mutant | / | / | / | / | / | / | 2.496 | 1.582–3.937 | 8.4E−05 | 0.995 | 0.600–1.650 | 0.983 |
| 1p19q_status | Non_codel vs. Codel | / | / | / | / | / | / | 6.554 | 3.358–12.790 | 3.6E−08 | 4.566 | 2.215–9.414 | 0.0000388 |
| Radiation therapy | Yes vs. No | 1.996 | 1.278–3.118 | 0.00236 | 0.814 | 0.488–1.358 | 0.43 | 0.475 | 0.262–0.861 | 0.0141 | 0.524 | 0.277–0.990 | 0.046 |

## GO and KEGG analysis of screened genes interacted with three-gene signature

For the "biological processes" (BP), chromosome segregation, nuclear division, mitotic nuclear division, organelle fission, mitotic sister chromatid segregation were the commonly enriched categories (Fig. 10A). For the "cellular component" (CC), the enriched categories were correlated with condensed chromosome, chromosome/centromeric region, chromosomal region, kinetochore, condensed chromosome/centromeric region

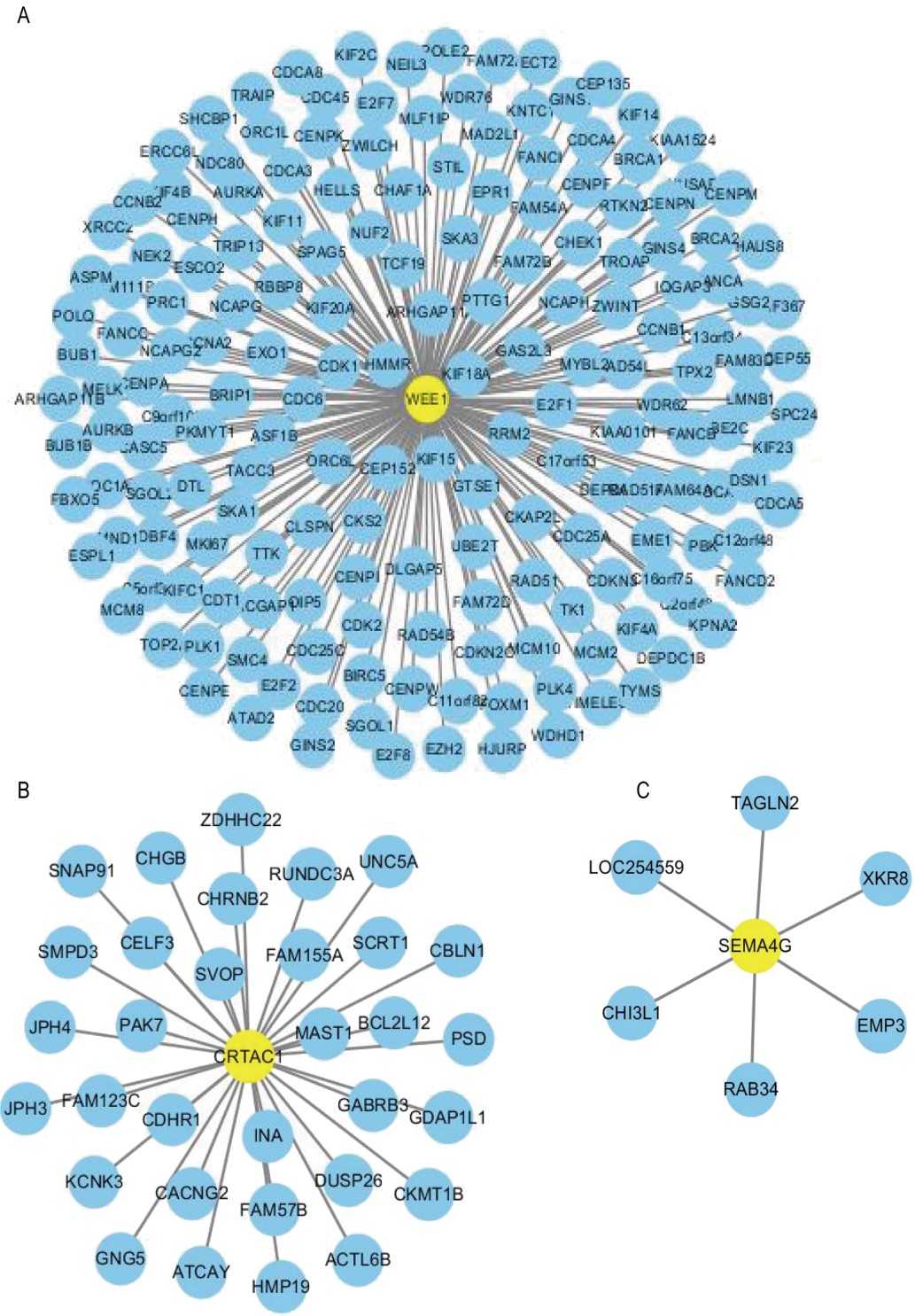

**Figure 9 Co-expression network of the three-gene signature.** The co-expression networks of WEE1 (A), CRTAC1 (B) and SEMA4G (C) are shown. Yellow nodes show key genes and blue nodes are genes which co-expressed with the key genes.

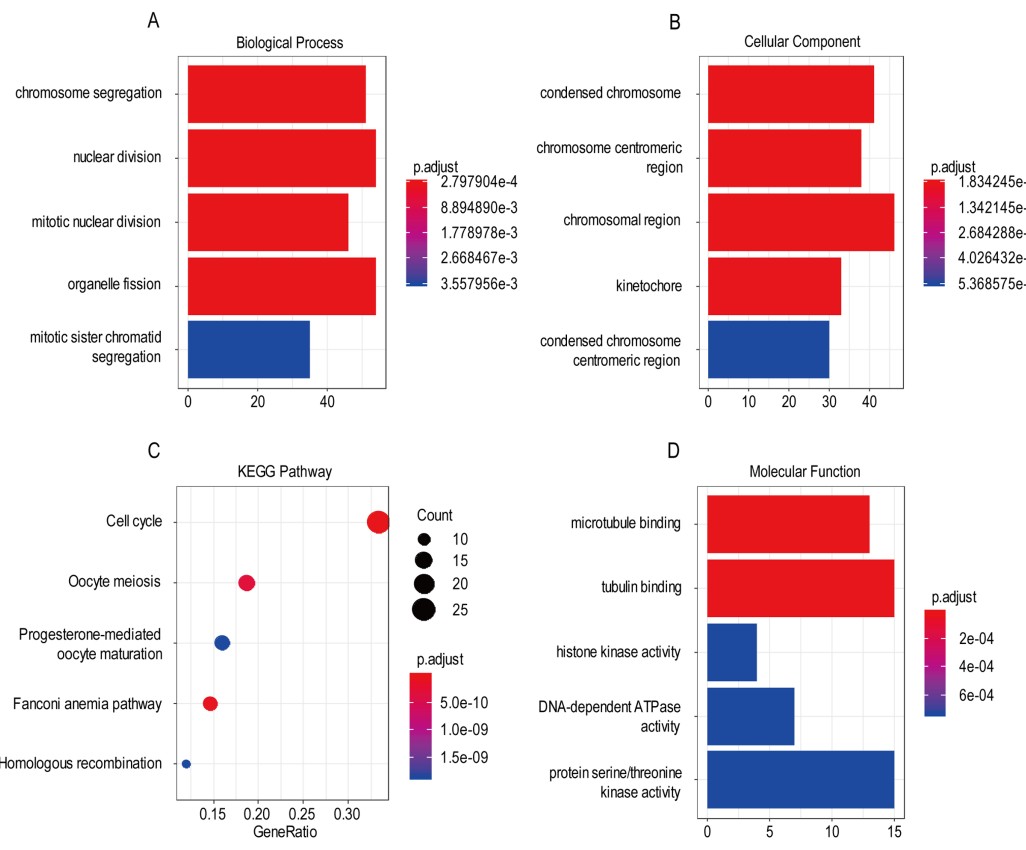

**Figure 10 The most significantly enriched GO annotations and KEGG pathways of co-expressed genes. The length of the bars and the size of the dots represents the numbers of genes, and the color of the bars/dots corresponds to the *p*-value according to legend.** (A) Top 5 significantly enriched biological process. (B) Top 5 significantly enriched cellular component. (C) Top 5 significantly enriched KEGG pathways. (D) Top 5 significantly enriched molecular function.

(Fig. 10B). For the "molecular function" (MF), these screened genes mainly enriched in microtubule binding, tubulin binding, histone kinase activity, DNA-dependent ATPase activity, protein serine/threonine kinase activity (Fig. 10D). KEGG pathway enrichment analysis suggested that cell cycle was the most important pathway for these selected genes. The following pathway also involved many screened genes, including, oocyte meiosis, progesterone-mediated oocyte maturation, fanconi anemia pathway, homologous recombination (Fig. 10C). Additionally, For the GO analysis, these three co-expression gene modules (yellow, green, black) enriched results can been seen in Figs. S2 and S3.

## A comparison between our and other models

Recently, *Zeng et al. (2018)* reported a model containing three genes (EMP3, GSX2, EMILIN3) based on integrative analysis of DNA methylation and gene expression in TCGA dataset. *Zhang et al. (2019a)* also reported a 4-gene (EMP3, GNG12, KIF2C, IFI44) prognostic signature based on genes encodes by chr1p/19q. To compare the prognostic values of our prognostic signature and their model, we performed time-dependent ROC curve analysis in our model and other models based on the risk score calculated by the

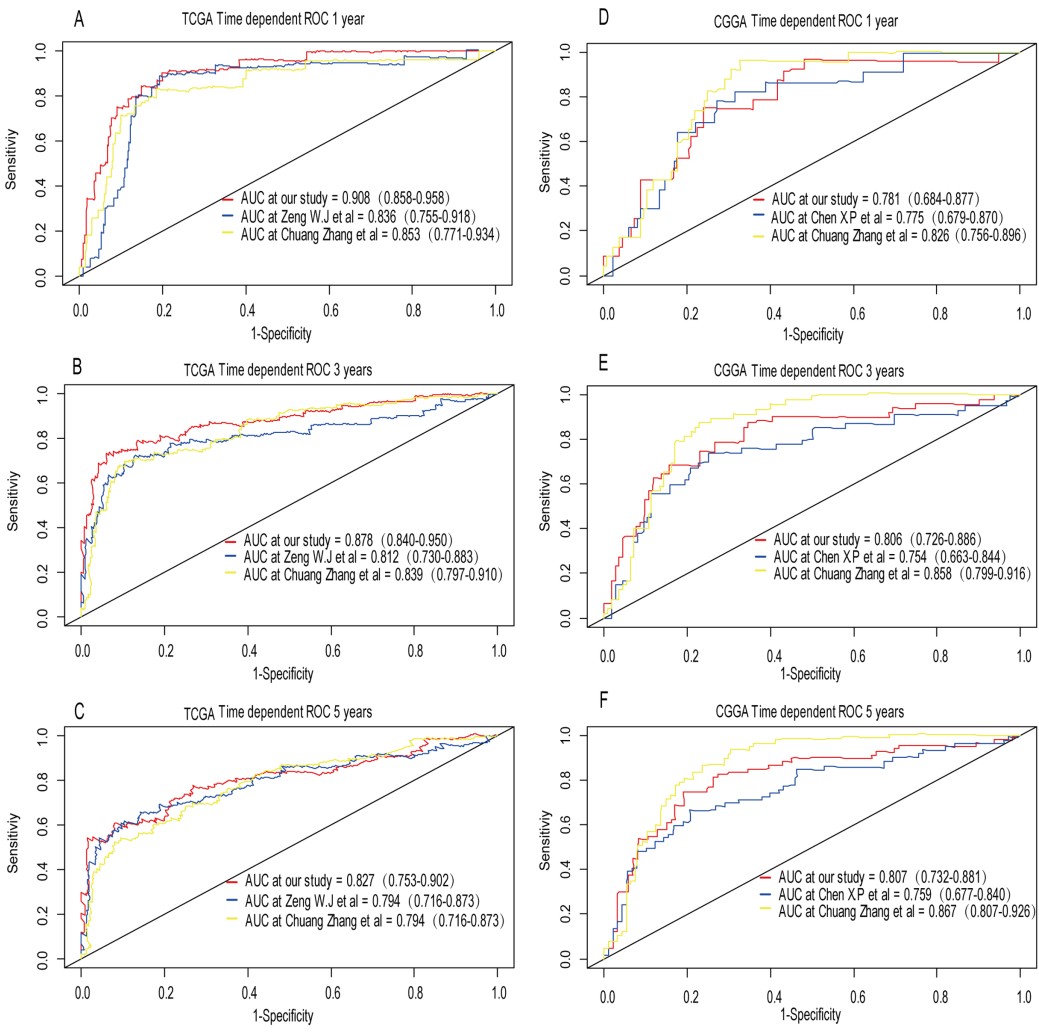

**Figure 11 Comparison of our 3-gene model and other literature models.** The-dependent ROC analysis was performed to compare the three models in predicting 1-year, 3-year, and 5-year overall survival in TCGA dataset (A–C) and CGGA dataset (D–F). 

regression coefficients which obtained by themselves and the expression level of members in their signature were shown in both TCGA and CGGA dataset. The results exhibited that our model displayed a satisfactory predictive value in predicting 1-, 3- and 5-year overall survival compared to other models (Figs. 11A–11F). In other words, our 3-gene model had a satisfactory efficiency in predicting both short- and long-term prognosis.

## DISCUSSION

From the perspective of traditional pathology, the diagnosis of low-grade glioma depends on pathological type and pathological grade. With the development of sequencing technology, molecular biomarkers for the diagnosis of LGG have attracted widespread attention (*Cancer Genome Atlas Research Network et al., 2015*). Prognostic factors for the low-grade glioma that are well known include IDH mutations (*Batsios et al., 2019*; *Ye et al., 2019*), 1p/19q co-deficiency (*Zhang et al., 2019a*), ATRX mutation (*Ren et al.,*

2019), TERT promoter mutations (*Chan et al., 2015*), CIC loss (*Sahm et al., 2012*), FUBP1 loss (*Sahm et al., 2012*) and PTEN loss (*Sabha et al., 2014*) and the above prognostic marker contribute to clinicians to understand the mechanism of low-grade gliomas. The complex pathogenesis of LGG encourages us to explore more prognostic markers for further understand it and develop an efficient treatment.

In this study, we identified three genes that were closely correlated with LGG prognosis. Considering the differentially expressed genes (DEGs) between tumor and normal tissue might not be associated with survival (*Liu et al., 2019*) and the univariate survival modeling can be performed in rbsurv package, the robust likelihood-based survival model was performed using the rbsurv package in R as the first step instead of screening for DEGs and conducting the univariate Cox regression. LASSO and Cox proportional hazard regression model are widely used to generate prognostic genes in the context of high dimensional data, thus were adopted in subsequent analysis. Compared to a single predictive biomarker, integrating multiple biomarkers into a signature is believed to be more predictive. The risk score calculated by the risk model was considered to have good predictive capabilities and was demonstrated to be an independent prognostic factor after adjusting the effects of age, sex, tumor grade, molecular therapy and radiation therapy. The risk score was confirmed to be effective in different age groups, gender and pathologic grade. Regardless of the training set or the validation set, the AUC value of 1-, 3- and 5-year was greater than 0.75. The pathologic grade and the risk level were confirmed to be independent prognostic factors both in training set and validation set.

In order to construct a co-expression network of the three genes, WGCNA was used in the training set. We found the survival related modules to which these three genes belong, and extracted the genes of these three modules to construct a co-expression network. Finally, 32 genes were discovered to be co-expressed with CRTAC1, 181 genes were co-expressed with WEE1, six genes with SEMA4G. The co-expression network of the three genes is visualized by Cytoscape in Fig. 9.

Based on the result of GO and KEGG enrichment analysis of these co-expression genes, "condensed chromosome" was the most significant enrichment in CC. Coincidentally, Rebecca et al. found that interference with chromatin condensation results in failure to fully activate DNA damage response (*Burgess et al., 2014*) and the DNA damage response triggers multiple cellular events including activation of DNA repair pathway, arrest of the cell cycle to allow time for repair, and, in certain cases, initiation of senescence or apoptosis programs (*Ciccia & Elledge, 2010*). For the BP category, chromosome segregation was the most enrichment and research has proven that chromosome instability contributes to the development of genetic heterogeneity in tumors and allows the outgrowth of tumorigenic cells with advantageous karyotypes (*Conde et al., 2017*). Regarding the MF category, microtubule binding was the most influential and the drug targeted microtubule was proven effective in glioma. For example, the drug EM011 functions by disrupting microtubule dynamics and modules several oncogenic mediators causing a decrease in cell viability, proliferation and migration/invasion in the astrocytoma cell lines (*Ajeawung, Joshi & Kamnasaran, 2013*). For KEGG pathway enrichment analysis, cell cycle was the most significant pathway. Stephen D has explained that

signaling pathway converge on the cell cycle machinery to regulate developmental genes and execute cell fate decisions (*Dalton, 2015*).

The three-gene signature provided a wealth of potential biological and therapeutic information about LGG.WEE1 (WEE1 G2 checkpoint kinase), located on the short arm of human chromosome 11 (11p15.4), encodes a nuclear protein, which is a tyrosine kinase belonging to the Ser/Thr family of protein kinases. The protein catalyzes the inhibitory tyrosine phosphorylation of CDC2/cyclin B kinase, and appears to coordinate the transition between DNA replication and mitosis by protecting the nucleus from cytoplasmically activated CDC2 kinase. WEE1 has been confirmed that its protein expression increases with malignancy grade (*Music et al., 2016*). Moreover, patients with high WEE1 expression had poor survival than did patients with low WEE1 expression in grade III gliomas (*Music et al., 2016*). CRTAC1, cartilage acidic protein 1, a novel human marker which allowed discrimination of human chondrocytes from osteoblasts and mesenchymal stem cells in culture can be divided into CRTAC1-A and CRTAC1-B two subtypes according to the last exon. Previous study found that inhibition of CRTAC1 reduces ultraviolet B irradiation induced-apoptosis through P38 mitogen-activated protein kinase and jun Amino-Terminal kinase pathway (*Ji et al., 2016*). It means that the relationship between the expression of CRTAC1 and apoptosis is positively correlated after ultraviolet B irradiation. To some extent, this is consistent with our finding that CRTAC1 high expression prolongs survival time in LGG patients. However, its detailed mechanism in LGG remains to be further explored. Semaphorins are a large family of conserved secreted and membrane associated proteins which possess a semaphoring (Sema) domain and a PSI domain in the N-terminal extracellular portion. Based on sequence and structural similarities, semaphorins are put into eight classes: invertebrates contain classes 1 and 2, viruses have class 8, and vertebrates contain class 3–7. Semaphorins serves as axon guidance ligands via multimeric receptor complexes, some containing plexin proteins. Semaphorins and Plexins are cognate ligand-receptor families that regulate important steps during nervous system development (*Maier et al., 2011*). A low-expression of SEMA4G was detected in colorectal cancer tissues compared with normal tissues. It means that SEMA4G might be a tumor suppressor gene related to colorectal cancer (*Wang et al., 2008*). However, little work has been done to elucidate the role of SEMA4G in glioma. Our study demonstrated that SEMA4G was significantly down-regulated in grade III patients compared to grade II and the high-expression of SEMA4G was associated with a good prognosis in LGG patients. Further work is needed to explore its functions in LGG. To sum up, the three-gene signature could predict LGG survival based on a risk score model. We firmly believed that these genes are potential prognostic markers or therapeutic targets for LGG patients. Nevertheless, the molecular mechanisms how the three-gene signature affected the prognosis of LGG patients should be further elucidated by a series of experiments.

## CONCLUSION

In conclusion, our study identified a 3-gene model that showed satisfactory performance in predicting short- and long-term survival of LGG patients compared to other models. Moreover, our finding provided new insights into the pathogenesis and prognosis of LGG.

## ACKNOWLEDGEMENTS

We thank editors for constructive comments on a previous version of this manuscript.

### Funding

This work was supported by the National Nature Science Foundation of China (No. 81801908) and the National Key Technology Research and Development Program of the Ministry of Science and Technology of China (No. 2014BAI04B01, No. 2014BAI04B02, and No. 2015BAI12B04). There was no additional external funding received for this study. The funders had no role in study design, data collection and analysis, decision to publish, or preparation of the manuscript.

### Grant Disclosures

The following grant information was disclosed by the authors:
National Nature Science Foundation of China: 81801908.
National Key Technology Research and Development Program of the Ministry of Science and Technology of China: 2014BAI04B01, 2014BAI04B02, and 2015BAI12B04.

### Competing Interests

The authors declare that they have no competing interests.

### Author Contributions

- Kai Xiao conceived and designed the experiments, performed the experiments, analyzed the data, prepared figures and/or tables, authored or reviewed drafts of the paper, approved the final draft.
- Qing Liu conceived and designed the experiments, authored or reviewed drafts of the paper, approved the final draft.
- Gang Peng analyzed the data, contributed reagents/materials/analysis tools, authored or reviewed drafts of the paper, approved the final draft.
- Jun Su performed the experiments, contributed reagents/materials/analysis tools, prepared figures and/or tables, approved the final draft.
- Chao-Ying Qin performed the experiments, authored or reviewed drafts of the paper, approved the final draft.
- Xiang-Yu Wang conceived and designed the experiments, analyzed the data, prepared figures and/or tables, authored or reviewed drafts of the paper, approved the final draft.

### DNA Deposition

The following information was supplied regarding the deposition of DNA sequences:
The RNAseq dataset is available at the UCSC Xena cohort: TCGA Lower Grade Glioma (LGG) and Chinese Glioma Genome Atlas.

## Data Availability

The raw data is available in the Supplemental Files, at UCSC Xena-TCGA Lower Grade Glioma (LGG)-gene expression RNAseq and the RNAseq dataset is available at the Chinese Glioma Genome Atlas.

## Supplemental Information

Supplemental information for this article can be found online at http://dx.doi.org/10.7717/peerj.8312#supplemental-information.

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
