# Peer review of "Identification and validation of a three-gene signature as a candidate prognostic biomarker for lower grade glioma"

_PeerJ, doi:10.7717/peerj.8312_

## Round 0.1 · original submission · Major Revisions

Please address the reviewers' comments.

Reviewer 1 ·

Basic reporting

1.1. Language
1.1.1. Line 24: please give the full form of WGCNA, since it is first mentioned here in the Abstract.
1.1.2. Line 47: As we known -> As we know
1.1.3. Line 52: clinically indicators -> clinical indicators; targeted therapeutic targets -> therapeutic targets
1.1.4. Line 73-75: This sentence is directly extracted from the abstract of the rbsurv paper by Cho et al. I would suggest rewriting the second half “and separates training and validation sets of samples for robustness” to avoid confusion on the terminology of training and validation. Something to the effect of “… and adopts a cross-validation approach for robustness”, for example.
1.1.5. Line 132: showed -> is shown
1.1.6. Line 166: Go -> GO
1.1.7. Line 176: Oocyte -> oocyte
1.1.8. Line 205: adopt to -> adopted in

1.2. Literature reference
1.2.1. Line 47-51: suggest adding references for molecular markers used in clinical tests of glioblastoma.
1.2.2. Line 56: Please put a citation for UCSC Xena (https://doi.org/10.1101/326470) and for CGGA (https://www.nature.com/articles/sdata201724).
1.2.3. Line 65: Please add appropriate citation or acknowledgement of TCGA. See https://www.cancer.gov/about-nci/organization/ccg/research/structural-genomics/tcga/using-tcga/citing-tcga
1.2.4. Line 83: Work of Ouyang et al. (2019) is cited here without much context. I would suggest changing the citation at the end of this line to a separate sentence, “The same approach was used to identify gene signatures for endometrial carcinoma (Ouyang et al. 2019).”
1.2.5. Line 85: The citation “(Wang, Wang et al. 2019)” seems out of place. Was it meant to go with “Ouyang et al. 2019” in the previous sentence, or with the “prognostic risk score” at line 88?
1.2.6. Line 107: Please cite the WGCNA R package paper (https://doi.org/10.1186/1471-2105-9-559)
1.2.7. Line 116: Please cite Cytoscape properly.
1.2.8. Line 118: Which tool did you use to perform the GO enrichment analysis? Please give a citation, and also indicate the multiple test correction method for the reported adjusted p-values.
1.2.9. Line 129: If the authors want to give a citation for “risk score”, it should come in the Methods section where “risk score” is first mentioned.
1.2.10. Line 248-250: please reference appropriate work at the end of this sentence.

1.3. Article structure, figures tables, and raw data:
1.3.1. Figure 1: I find the color scheme a little confusing or distracting. Is the purple color intended for the training process or for analyses done on the TCGA data? Does the blue line indicate chronicle order of the analyses rather than a step-wise dependency between analyses? From the figure, it is not clear which dataset the WGCNA was performed on.
1.3.2. Figure 3: (B) Please indicate whether subjects are ordered by risk score in the figure legend. (C) Which expression level metric is used in this heatmap?
1.3.3. Line 158-165: (1) This is a paragraph in the Results section, but it is written in a way that like it belongs to “Methods”. Please include more detailed results, such as the total number of modules detected, number of modules associated with vital status, and number of genes in each of these modules. It seems that a paragraph in the Discussion section (line 214-219) contains this information and should have been placed in the Results section. Please address this issue. (2) Please add figure title and legend for the supplemental figure.
1.3.4. The supplemental file “peerj-41271-TCGA-LGG_clinicalMatrix.tar.gz” and file “peerj-41271-TCGA-RNAseq-HiSeqV2.tar.gz” appear to be only gzipped rather than tarballs. Please match the file name suffix with the format.
1.3.5. Supplemental code:
1.3.5.1. Comments in the R code were written in Chinese. I would suggest adding comments in English to serve the general readers.
1.3.5.2. Code file peerj-41271-lastcodeCGGA.R
1.3.5.2.1. Line 65, it appears that the first column contains CGGI subject ID, but it’s renamed to “Gene_Name” here.
1.3.5.2.2. Line 92 and line 105, the data is saved to “TCGA...Rdata” instead of “CGGA...Rdata”.
1.3.5.3. Code file peerj-41271-TCGAcode.R line 165, where in the code was this file “TCGA-LGG-rbsurv_result.txt” generated? I was able to run the code up to this line, but not beyond.

Experimental design

2.1. In Table 1 (referenced at line 69), suggest adding a p-value column to show the composition similarity/difference between the training and validation sets by individual clinical variables
2.2. Line 107: Please give more detailed criteria for filtering genes and samples and give the final number of genes and samples that WGCNA was performed on.
2.3. Line 125-126: “We further reduced the dimensionality of these high- dimensional data.” Which analysis does this sentence refer to?
2.4. By “modules associated with the three genes” (line 115) and “genes interacted with those three genes” (line 162-163), do you mean the modules that contain the 3 genes as members and genes belong to such modules? If this is the case, please explicitly state so, since “associated” and “interact” are quite vague descriptions.
2.5. For the Gene Ontology analysis, are the 3 co-expression gene modules (yellow, green, black) enriched with similar or different GO terms?

Validity of the findings

3.1. Line 178-186: Which dataset was the model comparison performed on? It would be key to know whether there is data leakage in this comparison.

Reviewer 2 ·

Basic reporting

There are typos and some format issues in the text. Please correct them.

Experimental design

- Why do you need to reference Wang 2019 when describing the risk score on Line 129 in the first paragraph of Results? Isn't this the new results reported here? Using the estimated regression coefficients may not be the most accurate. You could build a classifier just using the 3 genes, and use the coefficients from the classifier.

Validity of the findings

- In Figures 4E, F and G, the p-values need to be adjusted for multiple hypothesis testing.
- When reporting significant results, p-values should also be included in the text in Results. e.g. Line 147.

Reviewer 3 ·

Basic reporting

In this manuscript, the authors give a three-gene signature for predicting the prognosis of patients with LGG and show that the signature outperformed two other gene signatures. The authors should provide more details about the methods for the identification of survival-related genes and the comparison with other models.

Experimental design

The authors should provide more details about the methods for the identification of survival-related genes. How to deal with missing data? How about the normalization of gene expression levels? What are the parameters used in the R packages?

Validity of the findings

More details should be provided about the comparison with other models. What genes were used in the models? Were DNA methylation biomarkers used?

Additional comments

1. Abstract background: the main ?? of our study
2. Abstract methods: what is WGCNA
3. Line 65, full name of TCGA is needed.
4. Table 1, some information cannot be shown correctly, such as age group and sex.
5. Line 65, the data level of the TCGA data should be provided.
6. Line 71, it is unclear how missing data are handled.
7. Line 128, the author should clarify if the three genes have been reported in the literature.

---

## Round 0.2 · Minor Revisions

Please address the comments from R1 and R3.

Reviewer 1 ·

Basic reporting

The authors have addressed my previous comments. One minor typo on line 259: To compared -> To compare.

Experimental design

no comment

Validity of the findings

Line 259-267: I am concerned about the conclusion drawn from this comparison.

First of all, although the AUC by the model from this work is higher, I suspect the difference is insignificant. I would suggest adding a confidence interval to each AUC.

More importantly, the 3-gene-signature model developed in this work was trained on the TCGA dataset. The comparison with other models was also performed on this training set. The slightly better performance may be a result of overfitting. I strongly suggest the authors (also) report the comparison using the CGGA data. Otherwise, I think the authors should absolutely tone down their conclusion that their “3-gene model had a better efficiency” and make a note that the comparison was done using the training data.

Reviewer 2 ·

Basic reporting

no comment

Experimental design

no comment

Validity of the findings

no comment

Reviewer 3 ·

Basic reporting

The authors have addressed my concerns in the revised version.

Minor comments:

Some symbols in the captions of Figures 5, 6, and 7 cannot be shown correctly.

Experimental design

No comment.

Validity of the findings

No comment.

Additional comments

No comment.

---

## Round 0.3 · accepted · Accept

Please address the minor comment from R1 in the proofing PDF.

Reviewer 1 ·

Basic reporting

no comment

Experimental design

no comment

Validity of the findings

no comment

Additional comments

I appreciate the authors' efforts addressing my previous comments. My concerns have been appropriately addressed.

A minor comment: the AUC confidence intervals in Figure 4 B&D are not mentioned in the text or the figure legend; please add details regarding the CIs in the figure legend.

Reviewer 3 ·

Basic reporting

The authors have addressed my concerns in the revised version. The current version of the manuscript can be accepted.

Experimental design

N/A

Validity of the findings

N/A

Additional comments

The authors have addressed my concerns in the revised version. The current version of the manuscript can be accepted.